# Lung Inflammatory Phenotype in Mice Deficient in Fibulin-2 and ADAMTS-12

**DOI:** 10.3390/ijms25042024

**Published:** 2024-02-07

**Authors:** Yamina Mohamedi, Tania Fontanil, José A. Vega, Teresa Cobo, Santiago Cal, Álvaro J. Obaya

**Affiliations:** 1Departamento de Bioquímica y Biología Molecular, Universidad de Oviedo, 33006 Oviedo, Spain; 2Departamento de Morfología y Biología Celular, Universidad de Oviedo, 33006 Oviedo, Spain; 3Facultad de Ciencias de la Salud, Universidad Autónoma de Chile, Providencia—Área Metropolinana, Santiago de Chile 7500912, Chile; 4Departamento de Cirugía y Especialidades Médico-Quirúrgicas, Universidad de Oviedo, 33006 Oviedo, Spain; 5Instituto Asturiano de Odontología (IAO), 33006 Oviedo, Spain; 6Departamento de Biología Funcional, Área de Fisiología, Universidad de Oviedo, 33006 Oviedo, Spain; 7Instituto Universitario de Oncología del Principado de Asturias (IUOPA), 33006 Oviedo, Spain

**Keywords:** ADAMTS-12, fibulin-2, lung inflammation, lung cancer, LPS, urethane

## Abstract

Interaction between extracellular matrix (ECM) components plays an important role in the regulation of cellular behavior and hence in tissue function. Consequently, characterization of new interactions within ECM opens the possibility of studying not only the functional but also the pathological consequences derived from those interactions. We have previously described the interaction between fibulin2 and ADAMTS-12 in vitro and the effects of that interaction using cellular models of cancer. Now, we generate a mouse deficient in both ECM components and evaluate functional consequences of their absence using different cancer and inflammation murine models. The main findings indicate that mice deficient in both fibulin2 and ADAMTS12 markedly increase the development of lung tumors following intraperitoneal urethane injections. Moreover, inflammatory phenotype is exacerbated in the lung after LPS treatment as can be inferred from the accumulation of active immune cells in lung parenchyma. Overall, our results suggest that protective effects in cancer or inflammation shown by fibulin2 and ADAMTS12 as interactive partners in vitro are also shown in a more realistic in vivo context.

## 1. Introduction

Chronic inflammation caused by infections or physicochemical agents is associated with approximately 20–25% of cancers [1,2]. In fact, the inflammatory response plays an important role in all cancer states: initiation, development, conversion to malignancy, invasion and metastasis [3]. The crosstalk between inflammation and cancer pathways is carried out, among others, by the production of cytokines and chemokines that attract several populations of immune cells, including macrophages, neutrophils, adipose cells, and lymphocytes, which infiltrate and become part of the tumor microenvironment [4]. In addition, two of the molecular pathways involved in inflammation, NF-kb and STAT3, are activated in most cancers [5]. Furthermore, the presence of inflammatory cells can stimulate the survival and proliferation of cancer cells [6]; and an incorrect response to inflammation can suppress the apoptotic processes of tumor cells, thus favoring the development of cancer [7].

Lung cancer is the most common cancer and deadliest human malignancy worldwide that causes approximately 1.5 million deaths each year (according to World Health Organization statistics). Tobacco and cigarettes are the first cause of lung cancer [8]. Cigarette smoke stimulates the recruitment of inflammatory cells, cell death, and protease production within the lung tissue [9,10]. Recent evidence demonstrates that cigarette smoke induces an M2 phenotype in macrophages via the JAK2/STAT3 pathway [11]. Also, cancer-associated fibroblasts (CAFs) play a prominent role within the reactive stroma, are responsible for the deposition of elevated amounts of extracellular matrix (ECM) and, at the same time, cooperate with and secrete a variety of enzymes that degrade the ECM [12,13]. Any new contribution to clarifying the role of ECM components and its interactions in lung physiology could serve as an incentive to seek new advances in the treatment of pulmonary pathological processes.

Fibulin-2, a member of the fibulin family of ECM proteins, is mainly localized between microfibrils and the elastin nucleus in elastic fibers. Not only does fibulin-2 have a structural function since it also participates in the organization of the extracellular matrix and tissue morphogenesis [14]. In general, fibulin-2 participates in cellular processes such as adhesion, migration and proliferation and has also been implicated in a dual role in cancer progression [15]. Thus, fibulin-2 could be considered as a tissue and plasma biomarker of breast tumors [16]. However, using a panel of breast tumor cell lines it has been shown that the loss of expression of fibulin-2 is related to a greater tumor progression, which suggests its role as a tumor suppressor [17]. A tumor suppressor role for fibulin-2 has also been observed in gastric cancer [18]. In contrast, in other different types of cancer, a promoter role has been defined for fibulin-2. As an example, such is the case of hepatocellular carcinoma and lung adenocarcinoma [19,20]. The absence of fibulin-2 does not have any effect in the development since deficient mice are viable, fertile and show no apparent anatomic defects [21]. In these animals, levels of fibulin-1 are elevated compared with wild type mice, suggesting that the absence of fibulin-2 could be functionally compensated by fibulin-1. Furthermore, the relationship between members of the fibulin and ADAMTS family of proteins has been described with implications in the pathological processes and modulation of cancer cells behavior [22,23,24]. Among the latest, fibulin-2/ADAMTS-12 interaction has been described although the implication of this relationship in physiological processes has not yet been well characterized [23].

The ADAMTS (A *D*isintegrin And *M*etalloproteinase with *T*hrombo*S*pondin domains) comprise a family of 19 extracellular human metalloproteinases containing a variable number of type-1 thrombospondin (TSP-1) domains at the carboxy-end region that provides specific binding properties to ECM molecules that makes them unique [25]. ADAMTSs are associated with a series of processes such as cell proliferation, adhesion, migration, angiogenesis and maturation of stroma cells, which would finally have an important impact on tumor biology [26]. In addition, the complexity of functions of ADAMTSs is also evident in pathological situations as occurs in cancer, displaying a dual role and being able to act as pro- and anti-tumoral proteins by mechanisms that have not been completely explored [26].

Since the discovery and cloning of the gene encoding for the ADAMTS12 metalloproteinase, a growing number of studies have shown the participation of this enzyme in physiological and pathological processes [27,28,29,30]. These include arthritic and inflammatory diseases [31], neuronal disorders [32] and cancer [33]. ADAMTS12 seems to be involved in the resolution of normal inflammatory processes so the absence of this protease increases the symptoms in diseases such as colitis and pancreatitis, as well as damages caused by septicemia in animals deficient in ADAMTS12 [34]. These mice are also especially susceptible to bronchial inflammation when exposed to certain allergens, showing an exacerbated increase in eosinophils both in bronchoalveolar lavage and in the lung tissue [35,36]. Regarding cancer, ADAMTS-12 was initially described to show an anti-tumor effect in Madin Darby canine kidney (MDCK) cells by inhibition of the Ras-MAPK signaling pathway [37]. In colorectal carcinomas, ADAMTS12 is downregulated by the gene promoter hypermethylation [38,39]. Bioinformatics studies and RNA interference techniques also supported a protective role of ADAMTS-12 in lung cancer [33]. By contrast, a protumor effect of ADAMTS-12 has also been described in ovarian cancer and in renal, gastric and esophageal squamous cell carcinomas [40,41,42]. ADAMTS12 could promote pro-tumor properties in breast cancer cell lines. Interestingly, the presence and interaction of fibulin-2 and ADAMTS-12 results in anti-tumor effects in breast cancer [23].

In the present work, we generate mice double-deficient in Fibulin-2 and ADAMTS-12. The absence of these two proteins did not cause apparent phenotypic alterations in terms of viability, fertility, or obvious major macroscopic abnormalities. However, we found that fibulin-2^-/-^ADAMTS12^-/-^ mice have inflammatory-like lesions in lung parenchyma, which become more evident after LPS treatment or in models of lung carcinogenesis, illustrating the relevance of the ADAMTS12 and fibulin-2 in lung homeostasis and pathologies.

## 2. Results

### 2.1. Double Fibulin-2 and ADAMTS-12-Deficient Mice Are Viable, Fertile and Show No Obvious Macroscopic Phenotype

ADAMTS-12-deficient mice [43] and fibulin-2-deficient mice kindly provided by Dr. Takesi Tsuda (Nemours Cardiac Center, Nemours/Alfred I. duPont Hospital for Children, 1600 Rockland Road, Wilmington, DE, USA) [21] were inter-crossed to generate the double knock-out mice for fibulin-2 and ADAMTS-12. Heterozygous mice from the F1 generation were then crossed to generate fibulin-2/ADAMTS-12-deficient mice. Mice genotypes were checked by PCR and Western blot (Figure 1). As shown in Figure 1a, the PCR products consisted of fragments of 330 bp (wild-type allele) and 270 bp (KO allele) in the case of *Adamts12* genotyping; and 480 bp (wild-type allele) and 440 base pairs (bp) (KO allele) in the case of *fibulin2* genotyping.

Despite fibulin-2 and ADAMTS-12 deficiency, these mutant mice developed normally, were fertile and did not have any obvious macroscopic phenotypes (Figure 1b). However, we performed an initial histopathological analysis of multiple tissues from fibulin-2/ADAMTS-12 adult animals on haematoxylin/eosin samples from the brain, lung, ovary, heart and colon that did not reveal any significant differences with the exception of lung samples (Appendix A). In this case, the double KO mice showed clear alterations in lung parenchyma compatible with inflammatory lesions (Figure 1b and Appendix A).

### 2.2. Analysis of Lung Cancer Susceptibility in Fibulin-2/ADAMTS-12-Deficient Mice

The dispensability of fibulin-2 and ADAMTS-12 for mouse development, growth and fertility opened the possibility of performing long-term studies in mice deficient in both proteins. Our experiments were aimed at analyzing the effects on lung tissue considering the previous inflammation phenotype (Figure 1b) and the susceptibility to lung cancer of ADAMTS-12 knock-out mice [33]. For this purpose, we chemically induced lung carcinoma following the intraperitoneal injection of urethane model to study the effects on fibulin-2/Adamts12 mutant mice. For this protocol, animals were treated with urethane and, after 20 weeks, the mice were sacrificed and the number and size of the lung tumors were quantified for all experimental groups (Figure 2 and Appendix A). The total number of lung tumors per mouse was higher in fibulin-2/ADAMTS-12 deficient mice than wild-type (five times higher) (Figure 2). The enrichment in the number of tumors was still significant after disaggregating tumors by their size (large (>400 µm), medium (200–400 µm) and small (<200 µm) tumors). The simultaneous absence of both ECM components could be in line with the antitumor function associated with each one separately.

### 2.3. Fibulin-2 and ADAMTS12 Deficiency Results in Increased Inflammation and Higher Susceptibility to LPS

Lung injury was assessed without LPS and 8 h after LPS administration. Mice lacking in fibulin-2 and ADAMTS-12 developed a more severe lung injury than their wild-type counterparts (Figure 3), as demonstrated by their higher histological scores also showing an enlargement of the distal airways and alveoli and the accumulation and extravasated blood cells in the lung of *fibulin2^-/-^/Adamts12^-/-^* animals. These results suggest an increased inflammatory response within the lung.

### 2.4. Increased Inflammation in Fibulin-2/Adamts12^-/-^ Mice in Lungs Is Associated with Macrophages, Neutrophils and Lymphocytes Accumulation

Examination of three different areas in the lung sections: zone 1 (central, peribronchiolar, and perivascular), zone 2 (intermediate part of the lung segment) and zone 3 (peripheral part of the lung segment) with an anti-MPO, anti-CD3 and anti-F4/80 antibodies revealed a marked increase in neutrophils, lymphocytes, and macrophages infiltration in *fibulin2^-/-^/Adamts12^-/-^*deficient animals (Figure 4A–C and Figure 5A–C).

The structure of the lung in WT mice is totally normal. F4/80 immunoreactive cells were observed below the basal membrane of the bronchioles with a very variable frequency. Additionally, the density of F4/80 positive within the bronchiolar epithelium varied among microscopic fields and animals (Figure 4B). In one section, a ganglion of the BALT (bronchial associated lymphoid tissue) whose germinal center contains abundant F4/80 positive cells was observed. The immunoreactivity for CD3 and MPO was negative (Figure 4A,C and Table 1). Conversely, animals deficient in fibulin-2 and ADAMTS-12 present a normal lung structure in the peripheral and intermediate segments. However, they also present a significant increase in BALT of peribronchial and perivascular localization that encompasses the blood vessels, and in their germinal centers there are abundant CD3 positive cells (Figure 4A). On the other hand, the density of F4/80 cells is increased in all sectors; in the alveoli, the F4/80 cells may correspond both with interstitial macrophages as type 2 pneumocytes (Figure 4B). We found positive MPO cells, at a very low density, in all segments of the lung (Figure 4C and Table 1).

After administration of a dose of 20 mg/Kg LPS, in control mice, the pattern of distribution of the antigens analyzed showed a higher density of immunoreactive cells for all of them and in all areas, but these were more evident in the intermediate and peripheral zones (Figure 5A–C). At the same time, in mice deficient in fibulin-2 and ADAMTS-12, after LPS treatment, an increase in the density of immunoreactive cells for the three antigens investigated in the three established zones was observed in general,. The most obvious elevations were produced for F4/80 and MPO in the intermediate and peripheral segments (Figure 5B,C and Table 1).

The semi-quantification of antigens CD3, F4/80 and MPO was carried out in such a way that the cell density was adjusted to the following scale: 0 means no positive cells; +: 1–5 positive cells per field; ++: 5–15 positive cells per field; +++: more than 20 positive cells per field (Table 1).

## 3. Discussion

Genetically modified mouse models show the intrinsic carcinogenic potency of inflammation and permit the molecular definition of several mechanisms whereby an incipient tumor takes advantage of an inflammatory microenvironment. During malignant transformation, cells obtain complex biological characteristics correlated with more efficient survival, invasion, metastasis, and the ability to evade the immune response [44,45,46]. The transformed cells communicate and alter the surrounding microenvironment consisting of extracellular matrix components, cytokines embedded in the ECM and the stromal cells [47,48]. Two families of proteins that form part of the ECM are fibulins and ADAMTSs. Different members of both families have been shown to interact and through their interaction modulate biological processes such as organogenesis or cancer development [22,23,24].

Differences in fibulin-2 expression in the tumoral processes have been described in breast, pancreatic and lung cancers, among others [17,20,49]. Elevated levels of fibulin-2 in the ECM of human lung adenocarcinomas, and in metastatic cells from a mouse model of a human lung adenocarcinoma underlined an important role of fibulin-2 as a promoter of lung cancer [20]. Part of the suggested mechanisms lies in the interaction with other ECM components like MUC4, a transmembrane type I glycoprotein. This interaction alters the integrity of the basal membrane and therefore promotes cell migration, invasion, and metastasis in these type of tumors [20,49]. In addition to the high expression of fibulin-2 on lung samples, a decrease in its expression in enriched epithelial cells of peripheral blood lymphocytes from the same patients has been observed. This loss of expression correlates with the metastatic potential of these cells and suggests that the fibulin-2 function might depend on the interaction with other molecules of the microenvironment [50]. Further regulation of fibulin-2 expression, through promoter methylation events, has been described in breast and lung cancers [51,52]. Cell-based assays showed the functional importance of fibulin-2 promoter methylation in the cell proliferation, migration, and invasion properties of lung cancer cell lines, and suggest a tumor-suppressive role for fibulin-2 in human cells [51]. Therefore, the dual role of fibulin-2 in cancer is clearly dependent on the cell origin (specie) and cell type and on the interaction with other ECM components. Fibulin-2 is also upregulated during wound repair processes and is involved in tissue remodeling that is required after tissue damage due to hypoxic stress [53,54]. Related to these processes, fibulin-2 may also promote fibroblast proliferation and migration and thus be involved in ECM deposition associated with idiopathic pulmonary fibrosis, a chronic progressive lung disease [55]. Interaction with ECM components like vitronectin and regulation of the anti-inflammatory cytokine TGF-β1 seem to be part of the mediated effects in proliferation, migration, and fibrosis [55]. In fact, an important crosstalk between fibulin-2 and the expression and activation of TFG-β1 seems to be occurring in fibrosis processes, cardiac remodeling, and ventricular dysfunction [53,54,55]. Thus, fibulin-2 can be associated with cancer progression with a role that clearly depends on the interactions with other components of the ECM. Among them, fibulin-2 interacts and can be degraded by members of the ADAMTS extracellular metalloproteases [23,24]. In particular, ADAMTS-12 interaction results in a protective function towards fibulin-2 degradation by ADAMTS-4 and ADAMTS-5 [24]. In addition, ADAMTS-12 can also be induced by the presence of TGF-β1 and has been described as participating in modulating tissue repair processes in the nervous system and as having a protective role in colon, liver and lung cancer where it is expressed in stromal fibroblasts [27,32,33,38,39]

Absence of ADAMTS-12 in mice, although proven to be involved in cancer and inflammatory processes, has not shown an effect in normal development. In particular, ADAMTS-12-deficient mice showed a susceptibility to developing breast and lung tumors [33,43]. In addition, they also showed significant impairments in recovery after pro-inflammatory challenges in the lung, pancreas, colon, and liver [27,34,35,36]. For its part, fibulin-2 deficiency also has no effect on normal mice development, likely due to fibulin-1 functional overlapping [21,56]. Yet, these mice showed important differences in recovery from several malignancies like myocardial infarction and bone fracture [54,57]. In the latest participation of fibulin-2 acts by modulation of the inflammatory response; levels of inflammatory cells are significantly lower in the spleen and thymus in *fibulin-2^-/-^* animals. Furthermore, fibulin-2 seems to act also as an inhibitor of BMSCs osteogenesis through inhibition of the Notch2 pathway [57,58].

Taking into account the interaction between fibulin-2 and ADAMTS-12 [23], we generated fibulin-2/ADAMTS-12-double-deficient mice and demonstrated that they represent a new and valuable in vivo model for the functional analysis of the interaction of these two proteins in inflammation and cancer processes. The lack of significant abnormalities in *fibulin2/Adamts12* mutant mice has facilitated studies aimed at evaluating their cancer susceptibility. Previous work in our laboratory has shown that the interaction between fibulin-2 and ADAMTS-12 promoted anti-tumor effects in breast cancer [23]. We focused our studies on the double knock-out mice in a lung cancer model since the *fibulin-2^-/-^/Adamts12^-/-^* mice presented significant lesions in the lung parenchyma. After application of a urethane protocol to induce lung carcinoma, we observed that mice deficient in fibulin-2 and ADAMTS12 showed a higher incidence of lung carcinomas than their corresponding wild-type littermates. These in vivo findings agree perfectly with the tumor suppression role already suggested for both proteins separately [33,51], although the urethane model did not suggest a protective role for fibulin-2 (Appendix A).

We also found that the loss of function of fibulin-2 and ADAMTS12 enhances mouse susceptibility to the inflammatory process. Thus, mutant mice exhibited an increase in damage after inflammation in the lungs during endotoxic shock. Histological analysis of the lungs revealed that the *fibulin2^-/-^/Adamts12^-/-^* mice show an increase in inflammatory infiltrates, interalveolar flooding, and in some cases a massive disruption of the lung structure. Immunocytochemical studies confirm and extend these differences; mice deficient in fibulin-2 and ADAMTS-12 show a greater accumulation of lymphocytes and neutrophils, but especially of macrophages, compared with control mice. These differences continue after LPS treatment, in which the two genotypes increase the reactivity of the CD3, F4/80 and MPO antigens, more specifically in the peripheral and intermediate zones.

In summary, the generation of mice deficient in fibulin-2 and ADAMTS-12 contributed to the in vivo validation that the interaction between these two proteins has an antitumor role in lung cancer and has an effect in modulating the immune response after lung injury. Therefore, although the absence of fibulin-2 and ADAMTS-12 is not essential for life, they play an important role in the complexity of processes such as tumor progression and inflammation.

## 4. Materials and Methods

### 4.1. Animals

Adamts12^-/-^ and fibulin-2^-/-^ mice were previously described [21,43]. Adamts12^-/-^ mice were repetitively crossbreed with fibulin2^-/-^ mice (kindly provided by Dr. Takeshi Tsuda, Nemours Cardiac Center, Wilmington, USA), to generate the double mutant mice fibulin-2^-/-^/Adamts12^-/-^. Genotyping was performed by PCR with DNA extracted from mice tails using the following oligonucleotides: 5′-GCTCGGTCAGGTGGCAGTGTA-3′, 5′-TTGTGTAGCGCCAAGTGCCCA-3′ and 5′-CCATTGTGAACGAGGTCCAGG-3′ for fibulin-2 and 5′-GGATGGTGTTGCCCACATTCA-3′, 5′-AGGCCAACTTGTGTAGCGCCAA-3′ and 5′-AGAGAGCATGCATGGGACAGA-3′ for ADAMTS-12. PCR was performed during 35 cycles of denaturation (94 °C, 30 s), annealing (62 °C, 30 s) and extension (72 °C, 1 min). The expected PCR products were DNA fragments of 440 base pairs (bp) (knock-out allele) and 480 bp (wild-type allele) in the case of fibulin-2 genotyping; and 270 bp (knock-out allele) and 330 bp (wild-type allele) in the cases of ADAMTS12 genotyping.

Mice were housed under specific pathogen-free conditions in 12/12 light/dark regime and fed ad libitum. All the experiments were performed following the guidelines of the Committee on Animal Experimentation of the Universidad de Oviedo and approved by the Consejería de Desarrollo Rural y Recursos Naturales del Principado de Asturias, Spain (authorization code: PROAE 02/2019).

### 4.2. Lung Carcinogenesis

To induce lung carcinogenesis, 8-week-old mice were intraperitoneally injected with 1mg/g body weight of Urethane (U-2500 Sigma-Aldrich, Sigma-Aldrich, St. Louis, MO, USA) [59], This protocol consists of two doses separated by 48 h and seven more doses on a weekly basis. After 4–6 months following the first urethane injection, the mice were sacrificed, and their lungs were fixed in 4% paraformaldehyde and paraffin-embedded. Serial sections every 100 μm were stained with hematoxylin/eosin for morphological examination by experienced pathologists. Tumors were quantified and classified by diameter measurement: large (>400 μm), medium (200–400 μm), and small (<200 μm) tumors.

### 4.3. Lung Inflammation by LPS Treatment

Eight-week-old mice were intraperitoneally injected with 20 mg/Kg body weight of lipopolysaccharide (LPS, from *E. coli* 055: B5, Sigma-Aldrich, St. Louis, MO, USA) and 8 h later were euthanized and their lungs removed for histological analysis. An inflammation score was obtained from a microscopic evaluation of standardly haematoxylin/eosin stained lungs with a semiquantitative scale ranging from 0 to 4: 0, normal lung; 1, capillary congestion; 2, alveolar wall thickening and inflammatory infiltrates; 3, intralveolar flooding; 4, massive disruption of the lung structure [60].

### 4.4. Structural and Immunohistochemical Analysis

To perform histological analysis immediately after extraction, tissue samples were fixed with 4% formaldehyde for 24 h, then washed with 70% ethanol and embedded in paraffin. Sections were standardly haematoxylin/eosin-stained and analyzed by an expert pathologist.

For the immunostaining of lung samples, we used the primary antibodies myeloperoxidase (MPO, A0398, Dako, Santa Clara, CA, USA) (1:500), CD3 (ab5690, Abcam, Cambridge, UK) (1:75) and F4/80 (sc-71086, Santa Cruz Technologies, Dallas, TX, USA) (1:200) for neutrophils, lymphocytes and macrophages, respectively. Sections were examined using a Nikon Eclipse E400 microscope and images were acquired with a Nikon DS-Sil camera (Nikon, Tokyo, Japan). Briefly, lungs were fixed in 4% formalin for 24 h. After fixation, samples were dehydrated and embedded in paraffin. Sections 4-μm thick were deparaffinized, rehydrated and then rinsed in PBS containing 1% Tween-20. Then, they were stained with haematoxylin and eosin for microscopy examination and adjacent sections were used for immunohistochemical labeling. Sections were incubated with primary antibodies and, after several washes, were incubated for 30 min with EnVision™+/HRP (Dako) and for 5 min with Liquid DAB (Dako). As immunostaining controls, representative sections were processed in the same way as described above using nonimmune rabbit or mouse sera instead of the primary antibodies or by omitting the primary antibodies during the incubation. For semi-quantitative analysis of inflammation in lungs, five consecutive sections (100 μm apart) were analyzed per each mouse and antigen under investigation. Each of the sections was divided into three zones: zone 1 (central, peribronchiolar and perivascular), zone 2 (intermediate part of the pulmonary segment) and zone 3 (peripheral part of the pulmonary segment). From each of the zones, five randomly selected fields were analyzed, using a 10× eyepiece and a 20× lens. Therefore, for each antigen and animal 75 fields were evaluated. Each field had a surface area of 1.5 mm^2^. Positive cell density was assessed as follows: 0: no positive cells; +: 1–5 positive cells per field; ++: 5–15 positive cells per field; +++: more than 20 positive cells per field.

### 4.5. Statistical Analysis

Data were analyzed using GraphPad Prism 7.0 Software and represented as means ± SE. Significant differences were determined with the Student–Welch *t*-test for parametric data and the Mann–Whitney test for non-parametric data. *p*-values < 0.05 were considered statistically significant (in the figures as: * *p* < 0.05, ** *p* < 0.01, *** *p* < 0.005).

## 5. Conclusions

ADAMTS-12 and fibulin-2 contribute to maintaining a lung protective environment in inflammatory and tumoral mice experimental models, resulting in their absence in a more profound pathological phenotype.

## Figures and Tables

**Figure 1 ijms-25-02024-f001:**
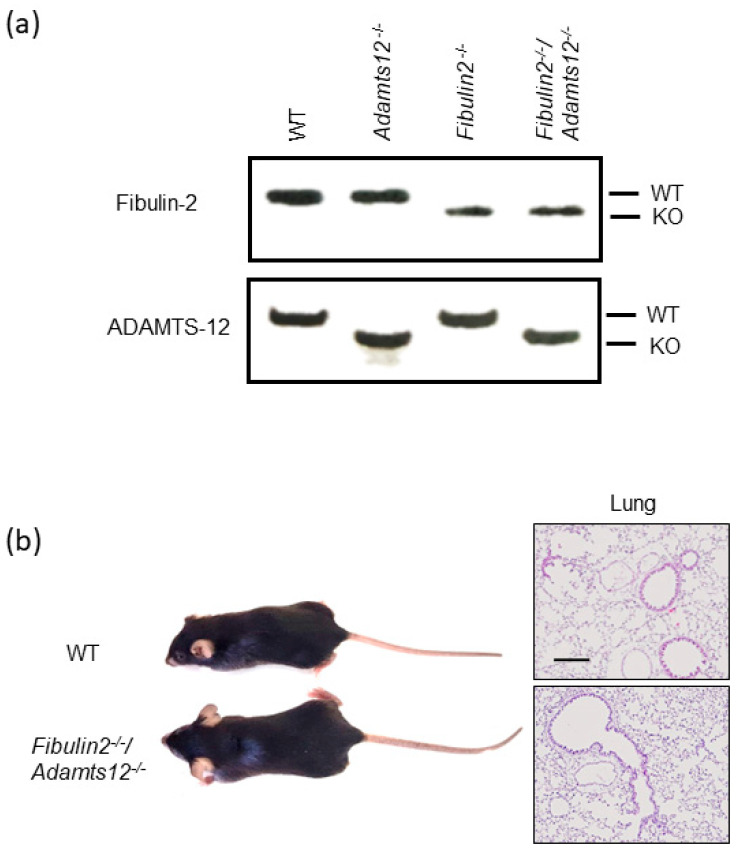
Generation of mice deficient in fibulin-2 and ADAMTS-12. (**a**) Example of PCR genotyping from tail biopsies of WT, *fibulin2^-/-^*, *Adamts12^-/-^* and *fibulin2^-/-^/Adamts12^-/-^* mice generated in C57BL/6 background. (**b**) Left, representative images of mice from wild-type (WT) and *fibulin2^-/-^*/*Adamts12^-/-^* mice. Right, representative images of haematoxylin/eosin-stained lungs from both genotypes. Scale bar: 200 µm.

**Figure 2 ijms-25-02024-f002:**
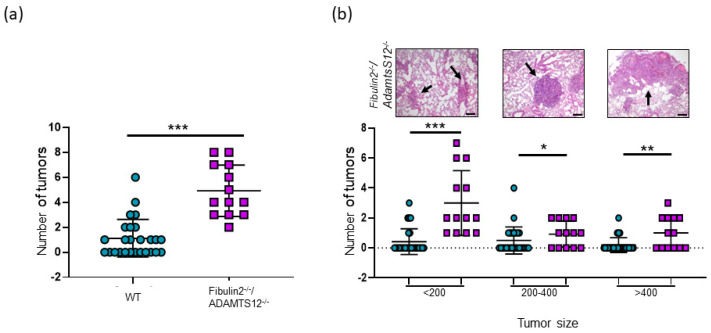
Lung cancer susceptibility in *fibulin2^-/-^/Adamts12^-/-^* mice. The left lung from each mouse was fixed and haematoxylin/eosin-stained. Each dot represents the total number of tumors quantified after serial section of the whole lung of an animal. (**a**) Number of total lung tumors per mouse after urethane treatment. (**b**) Graph, number of total tumors disaggregated by size. Circles, WT; squares, fibulin2^-/-^/Adamts12^-/-^ mice. Top: representative haematoxylin-eosin-stained tissue sections displaying lung carcinomas, Scale bar: 200 µm. *, *p* < 0.05; **, *p* < 0.01; *** *p* < 0.005.

**Figure 3 ijms-25-02024-f003:**
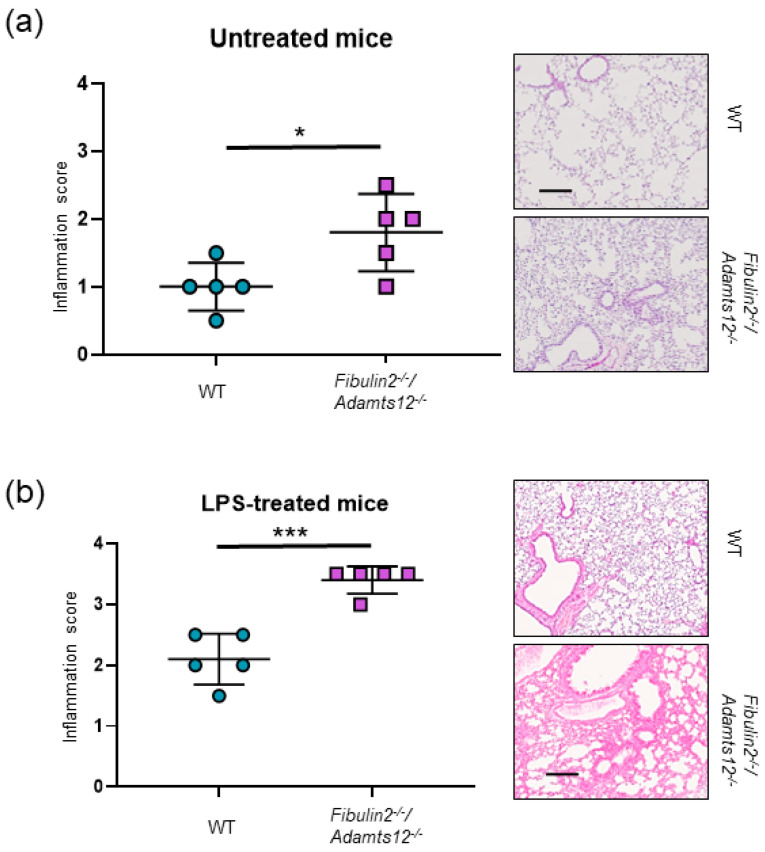
Increased susceptibility of *fibulin2^-/-^/Adamts12^-/-^* mice to LPS treatment. (**a**) Inflammation score (see M&M) of lung injury of untreated mice. (**b**) Inflammation score (see M&M) of lung injury of LPS-treated mice. Right in both panels, representative haematoxylin/eosin-stained images, scale bar: 200 µm. *, *p* < 0.05; *** *p* < 0.005.

**Figure 4 ijms-25-02024-f004:**
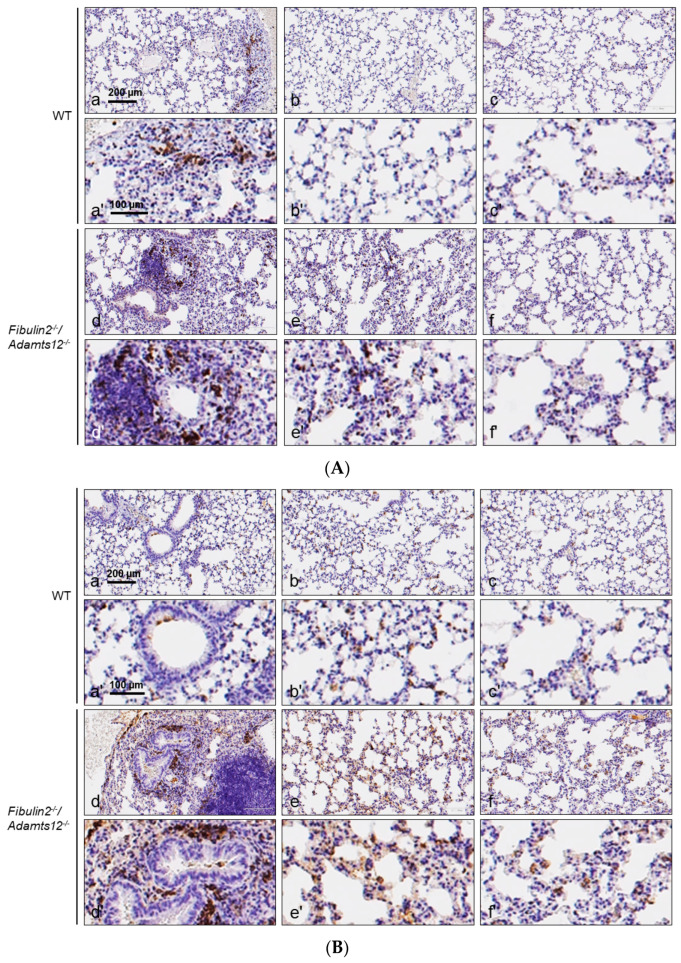
(**A**). **Normal lungs: immunostaining for CD3**. Accumulation of immune cells in lungs of mice deficient in fibulin-2 and ADAMTS-12. Representative images of CD3 (lymphocytes) staining in different areas of normal lungs of WT and *fibulin2^-/-^/Adamts12^-/-^*mice. Scale bar: identical value for (a–c) and (d–f), (a′–c′) and (d′–f′). Where possible, consecutive lung sections were used for immunostaining. Left column: peribronchial/perivascular areas; intermediate column: intermediate areas; right column: peripheral areas. (**B**). **Normal lungs: immunostaining for F4/80**. Accumulation of immune cells in lungs of mice deficient in fibulin-2 and ADAMTS-12. Representative images of F4/80 (macrophages) staining in different areas of normal lungs of WT and *fibulin2^-/-^/Adamts12^-/-^* mice. Scale bar: identical value for (a–c) and (d–f), (a′–c′) and (d′–f′). Where possible, consecutive lung sections were used for immunostaining. Left column: peribronchial/perivascular areas; intermediate column: intermediate areas; right column: peripheral areas. (**C**). **Normal lungs: immunostaining for MPO**. Accumulation of immune cells in lungs of mice deficient in fibulin-2 and ADAMTS-12. Representative images of MPO (active neutrophils) staining in different areas of normal lungs of WT and *fibulin2^-/-^/Adamts12^-/-^* mice. Scale bar: identical value for (a–c) and (d–f), (a′–c′) and (d′–f′). Where possible, consecutive lung sections were used for immunostaining. Left column: peribronchial/perivascular areas; intermediate column: intermediate areas; right column: peripheral areas.

**Figure 5 ijms-25-02024-f005:**
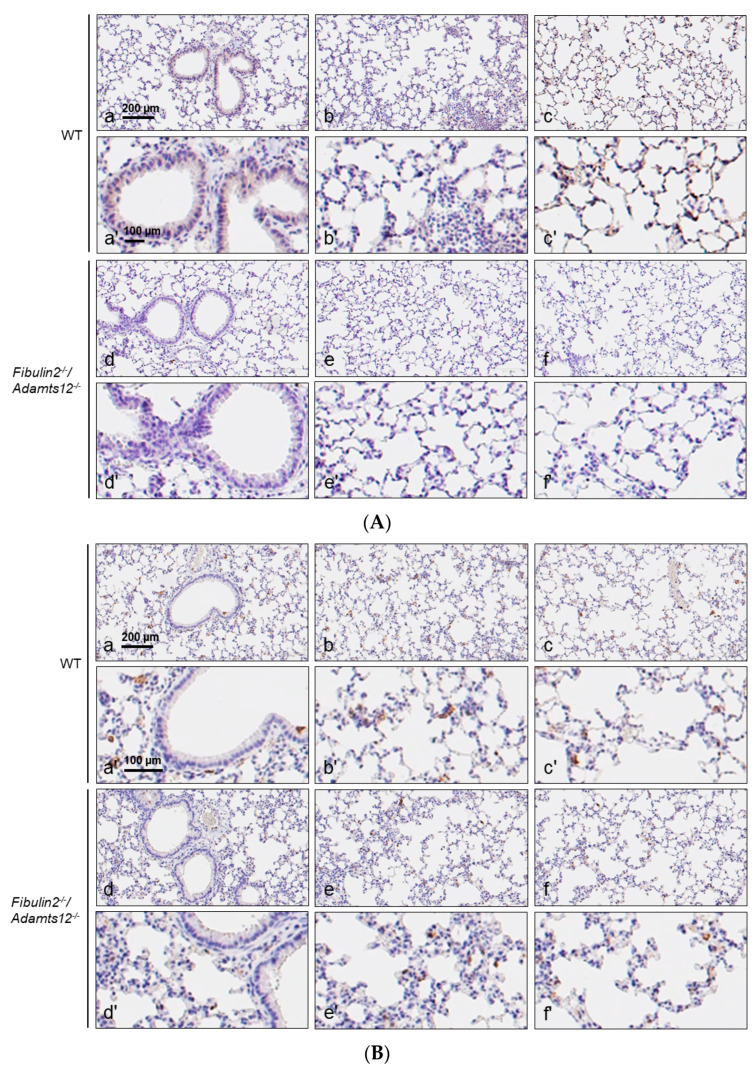
(**A**). **LPS-treated lungs: immunostaining for CD3**. Accumulation of immune cells in lungs 8 h later after LPS treatment in control and deficient in fibulin-2 and ADAMTS-12 mice. Representative images of CD3 (lymphocytes), staining in different areas of LPS-treated lungs of WT and *fibulin2^-/-^/Adamts12^-/-^* mice. Scale bar: identical value for (a–c) and (d–f), (a′–c′) and (d′–f′). Where possible, consecutive lung sections were used for immunostaining. Left column: peribronchial/perivascular areas; intermediate column: intermediate areas; right column: peripheral areas. (**B**). **LPS-treated lungs: immunostaining for F4/80.** Accumulation of immune cells in lungs 8 h after LPS treatment in control and deficient in fibulin-2 and ADAMTS-12 mice. Representative images of F4/80 (macrophages) staining in different areas of LPS-treated lungs of WT and *fibulin2^-/-^/Adamts12^-/-^* mice. Scale bar: identical value for (a–c) and (d–f), (a′–c′) and (d′–f′). Where possible, consecutive lung sections were used for immunostaining. Left column: peribronchial/perivascular areas; intermediate column: intermediate areas; right column: peripheral areas. (**C**). **LPS-treated lungs: immunostaining for MPO.** Accumulation of immune cells in lungs 8 h after LPS treatment in control and deficient in fibulin-2 and ADAMTS-12 mice. Representative images of MPO (active neutrophils) staining in different areas of LPS-treated lungs of WT and *fibulin2^-/-^/Adamts12^-/-^* mice. Scale bar: identical value for (a–c) and (d–f), (a′–c′) and (d′–f′). Where possible, consecutive lung sections were used for immunostaining. Left column: peribronchial/perivascular areas; intermediate column: intermediate areas; right column: peripheral areas.

**Table 1 ijms-25-02024-t001:** Semi-quantification of MPO, F4/80 and CD3 positive brown cells in normal lungs and lungs after an LPS treatment of WT and *fibulin2/Adamts12^-/-^* mice. Colors are used to identify each antigen. 0 means no positive cells; +: 1–5 positive cells per field; ++: 5–15 positive cells per field; +++: more than 20 positive cells per field.

	Parabronchial/Perivascular	Intermediate	Peripheral	Antigen
WT	0/+	0	0	CD3
	0/++	0/+	0/+	F4/80
	0	0	0	MPO
*Fibulin2* * ^-/-^ * */Adamts12* * ^-/-^ *	0/++	0	0	CD3
	++/+++	+/++	+/++	F4/80
	0/++	0/+	0/+	MPO
WT after LPS treatment	+/0	+/++	0/++	CD3
	0/++	+/++	+/++	F4/80
	+/+	+/++	0/++	MPO
*Fibulin2^-/-^/Adamts12^-/-^* after LPS treatment	0/+	0/+	0/+	CD3
	0/+	+/++	+/++	F4/80
	0/+	+/++	+/++	MPO

## Data Availability

The original contributions presented in the study are included in the article/Appendix A, further inquiries can be directed to the corresponding author.

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
