# Peer review of "Lung Inflammatory Phenotype in Mice Deficient in Fibulin-2 and ADAMTS-12"

_ijms, 2024, doi:10.3390/ijms25042024_

Round 1

Reviewer 1 Report

Comments and Suggestions for Authors

This study by Mohamedi Y et al., evaluated the role of fibulin2 and ADAMTS-12 in mice with LPS-induced lung inflammation and lung tumor model. The authors generated a double knockout mice model for the above functional gene. By performing immunohistology analysis and western blotting, authors found that these mice showed lung tumor growth. Additionally, these mice showed elevated inflammatory response, followed by LPS treatment.

While the presentation of double knockout mice is interesting, however there are multiple shortcomings underscoring this manuscript at its present level. Below are my concerns, which may authors to consider addressing them  in order to improve the manuscript.

1) Poor abstract: Abstract is the key to attain readers interest after the title. Abstract is weakened by generalized statements, therefore please strengthen the abstract by highlighting the findings.

2) Poor introduction: Since the title points to fibulin2 and ADAMTS12, the flow should start with the fibulin-2 and followed by ADAMTS12, I recommend authors to make these changes for a better flow. Additionally, please improve the introduction by citing recent findings in the field.   

3) Figure 1: Legend is incomplete. The authors presented the WT and double knockout (DKO) mouse images in 'b', however, the figure legend only describes IHC. Please add information on staining, i.e. H&E or eosin. IHC images are of low quality. line 132-replace black bar with 'scale bar'. 

4) Figure 2: Please modify graphs. Number of tumors per what? area or sections? how many sections were analyzed?  What is the 'n'. does the dot represent a section or a mouse?

5) Figure 3: please unify all figures to scatter plots with bars. The error bar is missing in 'b'. Its hard to understand the differences between figures. Provide a high quality IHC images.

6) Figures 4 & 5: A magnified area may help readers follow the figures; without this, all figures look similar. And, mainly, provide the quantification.

7) Table 1: Authors may consider moving this table to supplements.

8) Figure 6: Not considerable. Please provide convincing blots.

Major concerns: This data possesses an improper study design. The rationale and data discussion lead to confusion rather than understanding. I appreciate that the authors provided multiple stainings to prove that DKO showed increased inflammatory immune cell infiltration; however, the lack of mechanistic directions makes this study less interesting. The authors should greatly improve the quality of the presentation, strengthen the discussion, and propose a mechanism. This way, this manuscript may find a suitable place for publication.

I am regret to say that I am not in support of this manuscript at its present form.

Author Response

We are sending you the corrected manuscript according to all the suggestions received. First of all, we would like to thanks all the valuable comments of the reviewers that surely have improved the current version of our article. We have tried to address every points in an orderly manner a taking into account the comments of all of them with the answers point by point as follows;

  • Major changes and text addition is shown in red. All minor changes and typos are not indicated nor put in red.

1) Poor abstract: Abstract is the key to attain readers interest after the title.

Abstract is weakened by generalized statements, therefore please strengthen

the abstract by highlighting the findings.

                We have make changes in the abstract for highlighting our findings

2) Poor introduction: Since the title points to fibulin2 and ADAMTS12, the flow

should start with the fibulin-2 and followed by ADAMTS12, I recommend

authors to make these changes for a better flow. Additionally, please improve

the introduction by citing recent findings in the field.

                We have adjust the order of description of both proteins as recommended by the reviewer. We have also introduce some more recent references and also clarified the participation of both proteins in tumoral processes as suggested by other reviewer.

3) Figure 1: Legend is incomplete. The authors presented the WT and double

knockout (DKO) mouse images in 'b', however, the figure legend only

describes IHC. Please add information on staining, i.e. H&E or eosin. IHC

images are of low quality. line 132-replace black bar with 'scale bar'.

                We have redone the legend and improved images as suggested by the reviewer

4) Figure 2: Please modify graphs. Number of tumors per what? area or

sections? how many sections were analyzed? What is the 'n'. does the dot

represent a section or a mouse?

                We have redone the figures according to reviewer suggestions. In addition, we have clarify what represent each dot in the figure

5) Figure 3: please unify all figures to scatter plots with bars. The error bar is

missing in 'b'. Its hard to understand the differences between figures. Provide

a high quality IHC images.

                We have redone the figure to scatter plot wit bars. We have also improve the quality of the images.

6) Figures 4 & 5: A magnified area may help readers follow the figures;

without this, all figures look similar. And, mainly, provide the quantification.

                We have make the suggestions of the reviewer with changes in the figures by splitting each figure. In fact, we have divide each figure in three, one for each of the analyzed epitopes. Quantification was done and it is resumed in Table 1

7) Table 1: Authors may consider moving this table to supplements.

                We would like to keep this table since it summarizes and semiquantifies what is shown in figures 4A, 4B, 4C and 5A, 5B, 5C

8) Figure 6: Not considerable. Please provide convincing blots.

                We would like to keep this figure since it indicates that what is observed in IHC can also be followed at the molecular level.

Reviewer 2 Report

Comments and Suggestions for Authors

Comments:
The manuscript “Lung inflammatory phenotype in mice deficient in fibulin-2 and ADAMTS-12” by Obaya and their colleagues found that mice deficient in fibulin-2 and ADAMTS-12 has an anti-can cause lung cancer, which can potentially modulate the immune response after lung injury. Overall, this topic is not that interesting. There are a lot of typo errors in this manuscript. Just single technique (histology) as quantification technique in all those figures, which I think cannot well supported the conclusion. There are also so quite a few additional points of clarifications could potentially be addressed to further strengthen the manuscript.

1.     In line 79, the author stated ‘ADAMTS12 plays an anti-tumor effect in Madin Darby canine Kidney (MDCK) cells by inhibition of the Ras-MAPK signaling pathway. In colorectal carcinomas, ADAMTS12 is downregulated by gene promoter hypermethylation. By contrast, ADAMTS12 could promote pro-tumor properties in breast cancer cell lines in the absence of Fibulin-2.’ Here, MDCK is kidney cells, and also the author mentioned breast cancer cells, but all the following experiments and discussion are about lung cancer model, which is confused.

2.     Please ensure there is an in-text callout for Figure 1b.

3.     In line 121-127, the main text mention Figure 1c, but there is no Figure 1c in Figure 1. Please correct this.

4.     In line 124, the authors said there is no differences from brain, lung, kidney, colon in histology images (suggested adding those figures in SI).

5.     For student t test, please double check if *** is p<0.001 or p<0.005.

6.     In Figure 2a, caption ‘tumros’ should correct to ‘tumors’.

7.     In Figure 2, the mice number in your control group is much higher than your treated group. Then, this will lead to significant differences here. From the data shown here, for me, there are no significant differences between control and Fibulin2-/ADAMTS12- group. Just the authors add more mice number in control group, it causes significant differences here.

8.     In Figure, how the authors get the histological score, missing the explanation for that? What is the histological score for?

9.     What if there is just one gene deficient (for example, just Fibulin2- deficient or just ADAMTS12- deficient), does this will cause the lung tumor growth too?

1.  In Figure 4, it would be good to include the Flow cytometer to quantify the different percentage of immune cells.

Comments on the Quality of English Language

The authors need to work on the english and there are lots of typo errors in their manuscript. 

Author Response

We are sending you the corrected manuscript according to all the suggestions received. First of all, we would like to thanks all the valuable comments of the reviewers that surely have improved the current version of our article. We have tried to address every points in an orderly manner a taking into account the comments of all of them with the answers point by point as follows;

  • Major changes and text addition is shown in red. All minor changes and typos are not indicated nor put in red.

The manuscript “Lung inflammatory phenotype in mice deficient in fibulin-2 and ADAMTS-12” by

Obaya and their colleagues found that mice deficient in fibulin-2 and ADAMTS-12 has an antican

cause lung cancer, which can potentially modulate the immune response after lung injury.

Overall, this topic is not that interesting. There are a lot of typo errors in this manuscript. Just

single technique (histology) as quantification technique in all those figures, which I think cannot

well supported the conclusion. There are also so quite a few additional points of clarifications

could potentially be addressed to further strengthen the manuscript.

  1. In line 79, the author stated ‘ADAMTS12 plays an anti-tumor effect in Madin Darby canine Kidney

(MDCK) cells by inhibition of the Ras-MAPK signaling pathway. In colorectal carcinomas,

ADAMTS12 is downregulated by gene promoter hypermethylation. By contrast, ADAMTS12

could promote pro-tumor properties in breast cancer cell lines in the absence of Fibulin-2.’ Here,

MDCK is kidney cells, and also the author mentioned breast cancer cells, but all the following

experiments and discussion are about lung cancer model, which is confused.

                In this case, what we wanted is to indicate participation of ADAMTS-12 in tumoral processes. Thus, the focus of the paragraph was mainly the description of the general participation of this protein in cancer. The effect on MDCK was one of the initial observations and since then, ADAMTS-12 has been suggested to participate as a promoter as well as a suppressor in cancer. That is what we have tried to described in the paragraph. Modifications has been done in order to clarify this point.

  1. Please ensure there is an in-text callout for Figure 1b.

It has been done in the text

  1. In line 121-127, the main text mention Figure 1c, but there is no Figure 1c in Figure 1. Please

correct this.

                It has been corrected accordingly together with the previous point

  1. In line 124, the authors said there is no differences from brain, lung, kidney, colon in histology

images (suggested adding those figures in SI).

                We have introduce a supplementary figure 1 with some of the analyze histology images. We show the images of five different tissues a modified the text accordingly.

  1. For student t test, please double check if *** is p<0.001 or p<0.005.

                In our analysis it is p<0.005. We prefer to keep in this way although in some of our results it is even below 0.001.

  1. In Figure 2a, caption ‘tumros’ should correct to ‘tumors’.

                It has been corrected together with various typos we have seen in the main text

  1. In Figure 2, the mice number in your control group is much higher than your treated group. Then,

this will lead to significant differences here. From the data shown here, for me, there are no

significant differences between control and Fibulin2-/ADAMTS12- group. Just the authors add

more mice number in control group, it causes significant differences here.

                We have introduce a supplementary figure 2 in which the analysis was performed with even less animals and including single knock out animas. This was a different set of animals of what are shown in figure 2, The statistical difference stands even with this short number of animals. Anyway, the results for ADAMTS-12 were already published in Rabadan et al (2020).

  1. In Figure, how the authors get the histological score, missing the explanation for that? What is the

histological score for?

                We have change the term of histological score by inflammation score. The explanation of the inflammation score is described in Material and Methods.

  1. What if there is just one gene deficient (for example, just Fibulin2- deficient or just ADAMTS12-

deficient), does this will cause the lung tumor growth too?

                We have introduced and supplementary figure 2 with the results obtained when one gene was miising what is indicated also in the main text. In any case what happens with ADAMTS-12 was already published by Rabadan et al 2021

10 In Figure 4, it would be good to include the Flow cytometer to quantify the different percentage of

immune cells.

                We rather prefer to include the IHC images since we wanted to show what happens in the context of the tissue, to observe what parts were affected and, in which ones the accumulation of inflammatory cells occurred.

Reviewer 3 Report

Comments and Suggestions for Authors

Lung inflammatory phenotype in mice deficient in fibulin-2 and ADAMTS-12

Dear author and editor:

The article talked about the immunomodulatory role of fibulin-2 and ADAMTS-12. The article could be published after a revision. I have some comment on it:

·       The contrast of histological images in the figure 1 and all other  figures have to be adjusted and you have to write the scale bar on the figure.

·        The images in the figure 5 are not clear.

·        The table 1 could be organized in a better way.

·       The legend in the figure 6 could be written and explained in a better way and divide the immunoblott images to different lanes.

·       How do you explain the down regulation of tubulin in figure 6 in normal lung and LPS treated lung.

·       The conclusion should be written separately in the text.

Thank you very much, best regards

Author Response

We are sending you the corrected manuscript according to all the suggestions received. First of all, we would like to thanks all the valuable comments of the reviewers that surely have improved the current version of our article. We have tried to address every points in an orderly manner a taking into account the comments of all of them with the answers point by point as follows;

  • Major changes and text addition is shown in red. All minor changes and typos are not indicated nor put in red.

The article talked about the immunomodulatory role of fibulin-2and ADAMTS-12. The article could be published after a revision.I have some comment on it:

1 The contrast of histological images in the figure 1 and all other figures have to be adjusted and you have to write the scale baron the figure.

            We have redone the figures to improve quality and contrast and also to indicate “scale bar”.

2 The images in the figure 5 are not clear.

            As mention before, we have redone the figures in order to improve quality. In fact we have splitted figures 4 and 5 grouping the images for each epitope that was analyzed.

3 The table 1 could be organized in a better way.

            We would like to keep this table in the way it is since, in our opinion, it contain in summary what it is observed in figures 4 and 5

4 The legend in the figure 6 could be written and explained in abetter way and divide the immunoblott images to different lanes.

            We would like to keep this figure as it is with the blots put in columns for each treatment and in files for each western blot.

5 How do you explain the down regulation of tubulin in figure 6 innormal lung and LPS treated lung.

            We wouldn’t say it is a downregulation, it was much more a loading problem we had with the amount of sample we had at that moment and the need of keeping all the samples in the same blot. ·

The conclusion should be written separately in the text.

We added a conclusion paragraph at the end of material and methods section.

Round 2

Reviewer 1 Report

Comments and Suggestions for Authors

I appreciate the authors for revising the manuscript. However, this manuscript still contains multiple issues. 

1) Figure 3: legend indicates representative H&E images. How come similar tissue with the same staining showing a completely different staining pattern in color? 

2) Figure 4a, b: title of each panel should be readable; therefore, it should above the panel. scale bar not redable. There are multiple hidden white-colored lines appearing on the images. How come authors did not pay attention to these details while presenting representative images? this issue is persistent in this manuscript.

3) The author's response to my concerns about Western Blotts is inappropriate in my opinion. I assume that this data set calls into question the reliability if authors are unwilling to offer better blots and quantification. It could jeopardise the credibility of the entire work. 

Major: As I mentioned in my earlier report, this manuscript's depiction of the double knockout model is an intriguing feature. Nevertheless, it is clear that the authors failed to revise the manuscript and instead chose to disregard the reviewers' comments. 

I apologise to say that I am not in support of this manuscript.

Comments on the Quality of English Language

English-language editing is required.

Reviewer 2 Report

Comments and Suggestions for Authors

All the comments are addressed properly 

Author Response

Thank you for your help and comments

Reviewer 3 Report

Comments and Suggestions for Authors

The article could be published in the present form 

Author Response

Thank you for your help and comments

Round 3

Reviewer 1 Report

Comments and Suggestions for Authors

I appreciate authors for revising the manuscript. My concerns are in perspective of the broad readership, therefore, I emphasize on better, easy and reliable data presentation. Data visualization is equally important as the message it conveys. Poor quality data representation hinders readers interest. I apologize to say that I still have a few concerns that the authors can easily answer.

1) Fig 1 Lung IHC images were "boxed" (grey outlined), Fig 2, b-did not, while in Fig 3, some are boxed and others are not. And rest, all IHC images are boxed. Does the unboxed convey any meaning? Am I missing any details here?

2) Reg. Fig 3-staining variability I do not agree with authors response. Also, with scale bar generated by software. I would like to say that these details distract the reader's interest.

3) Fig4, 5 and 6: I clearly requested to make these labels readable. However, authors prefer not to change. Authors simply do not understand how difficult it is to read those images with the labeling that authors choose. For example> please carefully check Fig 4A> Top left image-no one can read the text there. and below middle panel, its hard to figure out that the marking is "e' but it looks like "e which its obviously utterly confusing. There are multiple such things. I am trying to make authors please be aware that this manuscript will be read by a broad readers from different fields. ALL FIGURES MUST BE CLEAR, SIMPLE, AND SELF-EXPLANATORY.

 4) Reg Western Blots: Blots were removed from the main figures and placed in supplementary figures. Why was this decision taken? Did I miss any details here? please clarify!!!. Even in supplements, blots were placed without any labeling. It's hard to read and understand without marking and proper labeling. I request that authors kindly pay attention for such details.